# It's Not Just a Phase: On Investigating Phase Transitions in Deep Learning-based Side-channel Analysis

## Abstract

Side-channel analysis (SCA) represents a realistic threat where the attacker can observe unintentional information to obtain secret data. Evaluation labs also use the same SCA techniques in the security certification process. The results in the last decade have shown that machine learning, especially deep learning, is an extremely powerful SCA approach, allowing the breaking of protected devices while achieving optimal attack performance. Unfortunately, deep learning operates as a black-box, making it less useful for security evaluators who must understand how attacks work to prevent them in the future. This work demonstrates that mechanistic interpretability can effectively scale to realistic scenarios where relevant information is sparse and well-defined interchange interventions to the input are impossible due to side-channel protections. Concretely, we reverse engineer the features the network learns during phase transitions, eventually retrieving secret masks, allowing us to move from black-box to white-box evaluation.

## 1. Introduction

Side-channel analysis (SCA) is a realistic security threat that consists of diverse methods that allow for the extraction and exploitation of unintentionally observable information of internally processed data (Kocher et al., 1999). SCA enables establishing a relationship between observable information and the internal state of a device under investigation. As such, it poses a major threat to devices that handle sensitive data like keys, private certificates, or intellectual property. In SCA, sensitive information gets extracted from a device by observing its physical characteristics (e.g., power consumption, timing).

Since 2016 (Maghrebi et al., 2016), deep learning-based side-channel analysis (DLSCA) has received significant attention from the research community (Picek et al., 2023). The main benefits of using deep learning (DL) over classical techniques are that assumptions for attacker capabilities can be relaxed and it leads to better attack performance. Thus, integration of these techniques into evaluation procedures has become standardized (Federal Office for Information Security (BSI), 2024). Note that (DL)SCAs are practical and demonstrated in real-world settings (Roche, 2024; Roche et al., 2021).

One of the main open challenges for black-box evaluations using DL is interpretability (Picek et al., 2023). A model that can extract the key suggests exploitable leakage but does not indicate how the network exploits what leakage. Notably, this does not allow the evaluator to provide any feedback beyond pass/fail, which complicates the cost-effective implementation of a solution. Indeed, understanding the mechanisms by which neural networks learn to exploit side-channel information can prove crucial for developing robust defenses against these attacks (Rijsdijk et al., 2022). Thus, several attempts have been made to understand network behavior. However, these approaches either focus only on input visualization (Masure et al., 2019; Hettwer et al., 2019), use more explainable model architectures (Yap et al., 2023; Yoshida et al., 2024) or require access to masking randomness (Zaid et al., 2023; Perin et al., 2022a).

While interpreting how neural networks perform computations is generally difficult, the algorithmic tasks performed in models trained on side-channel data are conceptually relatively simple. Learning to extract leakage information from masked implementations is similar to the toy models that learn group operations in the works on grokking (Power et al., 2022; Nanda et al., 2023a; Chughtai et al., 2023; Zhong et al., 2023). Concretely, for masked implementations, the computations on a sensitive value $s$ is split into $d$ secret shares $s = s_1 \cdot s_2 \cdots s_d$. Then, to learn to extract leakage from the side-channel signals, a neural network needs to combine leakage from each of these shares, often without the knowledge of individual shares even for the training set (Masure et al., 2023). This connection between side-channel and grokking models is further motivated by

[1]Anonymous Institution, Anonymous City, Anonymous Region, Anonymous Country. Correspondence to: Anonymous Author <anon.email@domain.com>.

Preliminary work. Under review by the International Conference on Machine Learning (ICML). Do not distribute.

Figure 1: The analysis approach used in this study broadly consists of three major steps. After the phase transitions are located using the PI metric, we plot logits to extract relevant features. Using these features, we plot the PCs of the activations and find the structure related to the leakage. Finally, we apply activation patching to reverse-engineer the masks.

the observation of Masure et al. that the learning curves for models trained against masked targets experience an 'initial plateau' (Section 5.2 of (Masure et al., 2023)). After a number of training steps where test loss does not improve, the models suddenly generalize to the test set and can extract the (sub)key.

These sudden increases in performance, i.e., phase transitions, raise the question of what the model is learning. Indeed, as some models for neural scaling predict neural networks learn in discrete steps (Michaud et al., 2023), we expect that investigating what is learned during these transitions will give a reasonable understanding of model behavior. Recent successful results of mechanistic interpretability (MI) investigating phase transitions in toy models (Nanda et al., 2023b; Simon et al., 2023) and even language models (Olsson et al., 2022) further motivate this direction.

From the point of view of MI, side-channel data provides an interesting test case. The data is often noisy, high-dimensional, characterized by subtle dependencies that are difficult to capture and interpret, and presents a real-world scenario. Additionally, the masks are hidden values which further complicates the application of MI as we cannot describe model behavior exhaustively concerning the concrete input features as in (Nanda et al., 2023b; Chughtai et al., 2023), or do (automated) input interventions to align with a causal model as in (Geiger et al., 2021; Conmy et al., 2023).

In this work, we aim to understand **what specific side-channel leakage the network is learning to exploit.** Concretely, we derive features from model outputs, find visual patterns (structures) that arise from principal components

(PCs) during phase transitions, and relate these to the physical leakage. As a practical consequence, we utilize this learned structure to extract input features, i.e., individual shares $s_i$, from model activations, providing a path to move from black-box to white-box evaluations. The overall analysis process is illustrated in Figure 1.

To summarize, our main contributions are:

- We explore the feasibility of applying MI in a challenging real-world setting where input interventions to features are not possible due to SCA countermeasures.
- By investigating the changes in model outputs during phase transitions, we find how networks combine leakage in DLSCA.
- We directly retrieve the secret share values leaked in a trace by applying activation patches[1] to intermediate layer activations across several targets.
- We provide more detailed insights into the specific physical leakage exploited by neural networks on widely used (DL)SCA benchmarks. Notably, we do this without assuming a priori mask knowledge (Zaid et al., 2023; Perin et al., 2022a) or requiring custom architectures (Yap et al., 2023; Yoshida et al., 2024).
- We find identical structures emerging during phase transitions for models trained on side-channel traces captured in different SCA domains and on different implementations, providing further evidence for the weak universality hypothesis (Chughtai et al., 2023).

---

[1]Activation patching is a technique from MI.

## 2. Side-channel Analysis

SCA consists of diverse methods that allow extracting and exploiting unintentionally observable information from internally processed data. SCA enables establishing a relationship between observable information and the internal state of a device under evaluation. In (physical) SCA, we attempt to extract secret information from side-channel traces, e.g., power/electromagnetic (EM) measurements, during the computation of a cryptographic algorithm. There, for $n$ encryptions[2] we collect $n$ traces of $m$ samples (features/points of interest) resulting in traces $\mathbf{X} = \{x^j, 1 \leq j \leq n\}$ where $x_j$ is a vector with $m$ points. Then, for each of these traces, key(s) $k^j$ and plaintexts $p^j$ allow us to generate a set of measurement labels. Let us consider the example of the NIST Advanced Encryption Standard (AES) cipher (Rijmen & Daemen, 2001), which is the algorithm of choice for most settings when encrypting information and is also the common target to explore in the research domain (Picek et al., 2023). AES is a byte-oriented cipher that operates in a number of rounds and where each of the rounds contains several operations. A common place to attack is after the S-box part, making the function of interest $IV = \texttt{S-box}(plaintext \oplus key)$. We can use the divide-and-conquer approach and consider attacking every key byte separately (as AES is byte-oriented); we denote the $i$-th byte of the key and plaintext as $k_i^j$ and $p_i^j$, respectively. The intermediate value then equals $IV^j = \texttt{S-box}[p_i^j \oplus k_i^j]$. Finally, when modeling the leakage, it is common to assume a certain behavior of how the device leaks, a concept known as the leakage model. Common leakage models include the Hamming weight (HW) leakage model[3], which assumes leakage is proportional to the number of ones in a byte, the least/most significant bit (LSB/MSB) that assumes the leakage happens in a single bit only, and the identity (ID) model that assumes that the leakage is proportional to the value at the output of the S-box. Note that since AES is byte-oriented, the S-box output contains 256 values, and the Hamming weight (distance) of those values can be between 0 and 8 (making a total of 9 values). While the ID leakage model is bijective, the HW/HD leakage models follow a binomial distribution. To assess the attack's effectiveness, it is common to consider how many guesses one needs to make before finding the correct key. As such, the fewer guesses, the better the attack (Picek et al., 2023). Another common metric to assess SCA performance is perceived information (PI), a lower bound for mutual information (Renauld et al., 2011). Masure et al. (Masure et al., 2020) showed that minimizing the negative log-likelihood is asymptoti-

cally equivalent to maximizing PI, making it relevant for the usage of deep learning and the metric we use in our analysis.

While many SCA variants exist, a common division is into direct and profiling attacks (Picek et al., 2023). Direct attacks assume a single device where the attacker uses statistical techniques (called distinguishers) to find the most likely keys. A common approach is the Correlation Power Analysis (CPA) (Brier et al., 2004). In profiled attacks, one assumes the attacker can access a copy of a device to be attacked. This copy is under the complete control of the attacker and is used to build a model of the device. Then, the attacker uses that model to attack a different (but similar) device. While the profiled attack is more complex due to the assumption of access to a copy of a device, it can be significantly more powerful than direct attacks. Indeed, provided that the model is well-built, one could need as little as a single trace from the device under attack to obtain the secret key. Direct attacks may need millions of traces to break a real-world target (Picek et al., 2023). One can easily observe a similarity between profiled attacks and the supervised machine learning paradigm (where building a model is training, and the attack is testing). Consequently, in the last decade and more, many machine (deep) learning algorithms have been tested in SCA.

As already stated, to protect against SCA, one commonly uses countermeasures that can be either hiding or masking. In both cases, the goal is to remove the correlation between the observed quality (traces) and secret information. Hiding countermeasures can happen in the amplitude domain by randomizing/smoothing the signal or by adding desynchronization/random delays in the time domain. Masking (Ishai et al., 2003), on the other hand, divides a secret variable into a number of shares such that one, to obtain the secret information, needs to know all the shares. For instance, consider a Boolean masking of a secret variable $x$. If we combine that secret variable with a random value $m$, we obtain a new variable $y$: $y = x \oplus m$. Then, to get information about $x$, one needs to know both $y$ and $m$. For further information about SCA, refer to (Standaert, September 2024).

## 3. Mechanistic Interpretability

Mechanistic interpretability (MI) aims to reverse engineer a neural network into human-understandable algorithms (Olah et al., 2020; Olah, 2022; Wang et al., 2023; Nanda et al., 2023b). This involves identifying "features" which are directions in internal representations that correspond to concepts, and "circuits" that are subgraphs within the network composed of interconnected neurons and their weights, representing meaningful computations.

Generally, the process in MI work is first to identify the features. Examples of features include low-level features such

---

[2]For simplicity, we mention only encryption; the process is analogous for decryption.

[3]Or the Hamming Distance (HD) leakage model that assumes the leakage is proportional to the number of transitions from zero to one and one to zero.

as curve or edge detector neurons in vision models (Olah et al., 2020), or more high-level features corresponding to the board state in toy models (Li et al., 2023; Nanda et al., 2023c). As features generally often correspond to linear directions in latent space, training linear probes (Alain & Bengio, 2017), i.e., small classifiers, is common for showing the presence of features in the latent space.

After finding features, the goal becomes to determine how these features relate to model outputs (or other features). Ideally, we can create a causal abstraction of the network behavior based on the feature descriptions (Geiger et al., 2021). Measuring causal effects involves intervening in the model activations by doing activation patches (Heimersheim & Nanda, 2024). Here, we replace (part of the) activations during a forward pass with saved activations from another forward pass corresponding to a different feature value to understand the effects on model outputs. This allows for measuring the impact of a specific feature or, eventually, verifying that the circuit is a (faithful) description of the model behavior.

## 4. Analysis Approach

The analysis process is shown in Figure 1 and detailed in this section. However, additional analysis and MI techniques might have been used depending on each specific dataset's observed behavior and findings. These additional steps and the reasons for them will be directly described within the experimental results (Section 5).

**Assumptions.** In (DL)SCA, the attack typically focuses on extracting a subkey (often a single key byte) of the secret key. Once one subkey is recovered, the target is effectively broken, as the attack can be repeated to recover the remaining subkeys. However, the effort for different subkeys can differ significantly (Perin et al., 2022b). Our analysis assumes the attack has already succeeded and the subkey has been recovered. This assumption allows us to label (test) traces by deriving the intermediate value (label) from the input (plaintext) and the key (remember $IV^j = \texttt{S-box}[p_i^j \oplus k_i^j]$). We do not assume we have access to mask values. **The goal is to understand the model's behavior and identify what information it extracts from the traces to make predictions.** Additionally, we aim to recover the masks used in the cryptographic algorithm, which enables us to recover the rest of the secret key with (significantly) less effort. Note that if the model's most likely subkey is incorrect, interpretability methods provide limited insights since the model presumably fails to learn the masks and features necessary for accurate key recovery.

**Logit Analysis.** Once we have a model that successfully recovers a subkey, our primary goal during initial exploratory testing is to understand the factors influencing the network's predictions. To achieve this, we analyze models at points directly after phase transitions and observe the changes to the model predictions. As phase transitions suggest significant changes in a neural network's behavior during training, they are shown to be useful for discerning features (Zhong et al., 2023). We examine the distributions of output logits for different classes, looking for clear separations between classes, indicating distinct patterns in the traces. We aggregate distribution for model outputs for traces that belong to each class and visualize them to identify commonly confused classes. Those insights enable us to formulate hypotheses about higher-level features influencing predictions. Opposed to other recent works that reverse engineer models, see, e.g., (Nanda et al., 2023b; Wang et al., 2023), where authors assign features to (or derive features from) model inputs, we rely on output logits as we do not have access to masking randomness. Additionally, the (physical) noise inherent to side-channel traces results in final model accuracies that can be only marginally above random guessing, making single trace predictions challenging to analyze from the MI perspective. Note that the analysis becomes easier in white-box SCA settings, where one would assume knowledge of all internal values during computation, including the masking randomness, see (Perin et al., 2022a).

**Activations Analysis.** After finding and testing initial hypotheses about the physical leakage used for classification, we can proceed to look at activations and how these relate to the predictions. As only a small number of the operations in each trace should be relevant for the classification, i.e., only leakage related to the target value is relevant, the number of relevant features should be small. Furthermore, during phase transitions, discernible structure emerges in PCs (Simon et al., 2023; Zhong et al., 2023).

As we expect structure to emerge in the first few PCs during phase transitions (Simon et al., 2023), we can plot the distribution of attack traces for features we derive from the logit analysis. Still, while we expect a specific structure would emerge in the first few PCs, some manual effort in determining the correct (number of) components and subdivisions might be necessary. However, in our case, we notice the structure generally emerges with up to the first four components. In ideal cases, we see clear divisions between groups of traces belonging to certain values. Even if this is not the case due to noise, some regions might contain more/less of certain groups, and the overall distribution should be tied to (noisy) physical leakage. After finding some structure, we should explain how it arises in terms of the physical leakage that is in the trace. For example, if there is a grid-like structure in the PCs, we could assume that (embeddings of) two secret shares correspond to the $x - y$ directions of the grid since we have two shares.[4]

---

[4]For higher $d$, the structure will be in higher dimensions.

**Reverse engineering masks with activation patching.**
When a structure is found, we need to verify that the hypothesized behavior is causally related to the model predictions in the expected way. We do this by fixing the directions that correspond to all but one share to a fixed value. If possible, we try to fix them to be 0 or some other value that allows for easy descriptions of the output based on the one varying share. Then, we observe how the model outputs relate to the final share. If the hypothesis about model behavior is correct, we can also directly derive the values for a secret share from these patched outputs. Finally, after deriving secret shares, we can use Signal-to-Noise Ratio (SNR)[5] to plot where in the trace these shares leak to derive which secret share is which, i.e., which of the two shares is the mask and which the masked S-box output. Note that this patching setup follows the activation patching method used in (Perin et al., 2022a) without requiring a priori knowledge of secret share values.

## 5. Experimental Results

This section presents results for three common public SCA targets - CHES_CTF, ESHARD, and ASCAD (Picek et al., 2023). The models are the Multilayer Perceptron (MLP) neural networks taken from (Perin et al., 2022a) for ES-HARD and ASCAD. For CHES_CTF, we directly train the ESHARD model without additional hyperparameter tuning. We focus on MLPs as these are generally sufficient for state-of-the-art (even optimal where the target is broken with a single attack measurement) performance in SCA (Perin et al., 2022b). The analyses given here should be similar for CNNs. Thus, the results on ESHARD and CHES_CTF datasets, which exhibit very similar leakage characteristics, both leaking mostly in the HW leakage model, follow the same process and similar findings. The ASCAD target has leakage biased toward the least significant bits, and a different MLP model is used, leading to slightly different findings and additional analysis required.

### 5.1. CHES_CTF Dataset

For the CHES_CTF target, we see in Figure 2 that there are two concrete increases in perceived information across training. The initial increase starts at epoch 15 and is completed around epoch 40. After another plateau in PI, there is a second increase between epochs 70-85, after which there are no more significant changes in PI.

As we aim to find what is learned during the phase transitions, we show both average logits for different classes and two main PCs in Figure 3. After the first phase transition,

---

[5]SNR measures the signal variance versus the noise variance. In SCA, a higher SNR indicates a stronger exploitable signal compared to noise, making it easier to extract sensitive information.

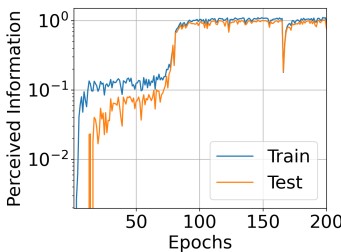

Figure 2: Evolution of Perceived Information for training and test traces of the CHES_CTF dataset.

at epoch 50, the predictions on the test set differentiate between high HW values and low HW values. When we use this information to plot PCs in the first layer (middle plot in Figure 3), we see that one diagonal corresponds to high HWs and the other to low HWs. This indicates that the HWs of both secret shares mask and masked S-box output leak in the HW leakage model and that these are the features that map onto the PCs. Further details are in Appendix E.

When looking at the logits after the second phase transition at epoch 100, Figure 3 shows that in addition to the high-low HW divide, the models also separate even-odd HWs. Plotting the same components but separating even-odd HWs shows a grid structure of even and odd points. In this grid, the number of changes in even-odd is about nine, corresponding to the nine possible HW values. The even-odd separation also clearly corresponds to learning the parity of a target value from HWs with Boolean masking (see Appendix E.2). This leads to the ability to learn the mask values the network uses for classification, as discussed next.

#### 5.1.1. ACTIVATION PATCHING

To validate that the PC embeddings are causally related to model outputs, we can fix one of the components and observe the effects on model outputs. An additional consideration is that when we fix the value of one of the Hamming weights to 0 (or 8), the output of the model should be the HW value of the other share (or 8-output if we fix the first to 8).[6] As such, if the PCs relate to mask values, we can patch one share to 0 (or 8) to retrieve the value of the other share.

To practically extract mask values, we fix the value of one PC to be (near) one of the corners of the grid we see in Figure 3. Then, we take the model outputs and check whether the predicted value changed as expected. As the model generally predicts HW values between 3-5 (because those

---

[6]Note that patching one share to be 0 to validate that the outputs become directly related to the other share has been done before in (Perin et al., 2022a) although by using knowledge of the masking randomness.

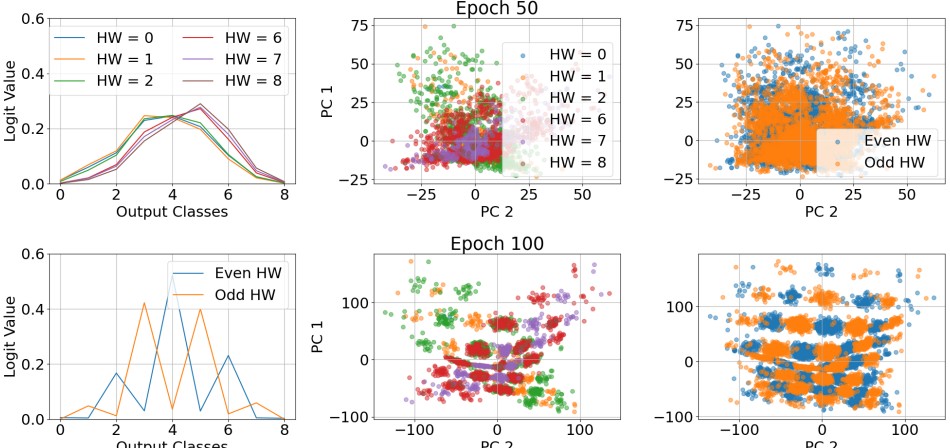

Figure 3: Logit analysis (first column) and activation analysis (remaining columns) from models at epoch 50 (top) and epoch 100 (bottom) for CHES_CTF. Legends for activation analysis are shared within columns. The difference in the number of points between the last two columns is due to not plotting the points for classes (HWs) 3, 4, and 5.

occur most), we sort each trace by the difference of logits for high (5-8) and low (0-3) HWs. Then, since we know the expected number of occurrences for each HW[7], we can take the first 1/256 values to be $HW = 0$, then the next 8/256 values for $HW = 1$, and so on.

The resulting mask and masked S-box distribution are shown in Figure 4. We can see that fixing values of certain PCs to extremes results in the model basing its predictions mainly on the other PC, as is expected when one of the shares is fixed to 0 (or 8). When we visualize the SNR for each share, we observe clear spikes corresponding to the usage of the leaking values. First, we see spikes related to the value of $r_m$, indicating the loading of the mask and some pre-processing before the encryption. Then later, we see leakage related to S-box$[p_i \oplus k_i] \oplus r_m$.

Due to the page limit, ESHARD results are in Appendix C. In summary, there is only one phase transition, which results in the ability of the model to distinguish high-low HWs. The results are qualitatively the same as for CHES_CTF.

### 5.2. ASCAD Dataset

Model behavior has been relatively straightforward in mapping to expected behavior with the HW leakage. However, for the ASCAD target (the main benchmark for DLSCA research since its introduction in 2018 (Benadjila et al., 2020)), the masking scheme is more complex, and the leakage is not directly tied to the HW of the full byte. As such, for this dataset, we additionally train linear probes for each bit of both the S-box input and output. One of the main

distinctions is also that for ASCAD, it is standard to use the S-box output values directly as class labels over transforming them into their HW values (see, e.g., (Benadjila et al., 2020; Perin et al., 2022b)).

Figure 5 shows a sudden transition to the positive PI from epochs 8-12, corresponding to increased probe accuracies for the input bits. Immediately after, PI still marginally increases until improvement stops at epoch 25. This increase is accompanied by the increasing probe accuracies for the two least significant S-box output bits. Indeed, the two least significant bits for both input and output clearly achieve far higher accuracies than other bits, which only marginally improve over random guessing (around 0.55).

Looking at the logits in Figure 6, at epoch 12, the values are distributed according to the two input bits. When we plot PCs to distinguish the values of these bits in Figure 6, we see an emerging structure in the first two PCs of the activations in the second layer corresponding to the combination of mask values by mapping these on certain axes. Note that the grid structure in both cases follows a $3 \times 3$ structure over the more ideal $4 \times 4$ if all four possible 2-bit values of the masks are perfectly distinguished. This is due to the physical leakage of two classes for the secret shares (mostly) overlapping, as shown in the rightmost two plots.

When we consider the logits at epoch 25 for the output bits,[8] the mean values are significantly higher. Additionally, the logits are spread out across fewer values. This aligns with the network's predictions, which now incorporate the information on the output bits. We also observe a visually similar

---

[7]If the mask values are uniformly distributed, which they generally are for the security properties to hold (Ishai et al., 2003).

[8]We fix the input bits to 00 to increase visibility, for a complete description, see Appendix F.1.

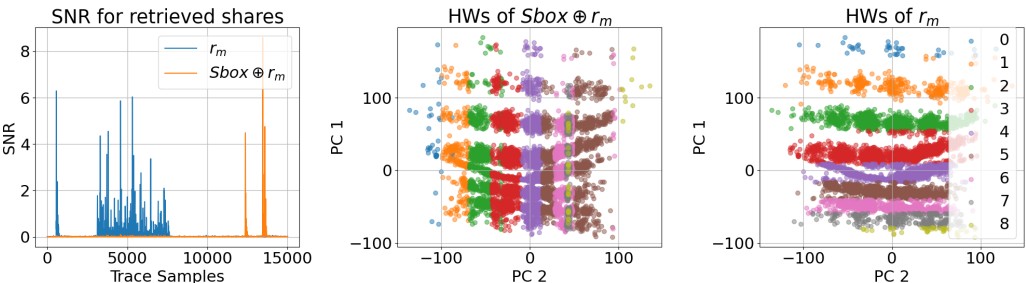

Figure 4: SNR plot and PC distributions for mask values using patching experiments for CHES_CTF. We set PC0 to -20 for both patching experiments, as that resulted in more apparent separation during manual testing.

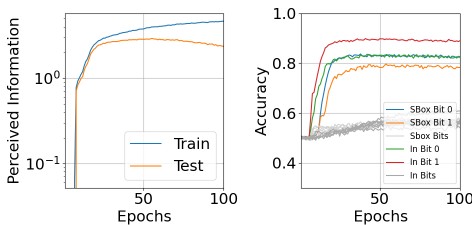

Figure 5: Evolution of Perceived Information and probe accuracies for bits during training for the ASCAD dataset.

structure to the grid at epoch 12 appearing in the 3rd and 4th PCs for the S-box output bits. The first two PCs remain related to the input bits as in epoch 12. Within the activation patching experiments for ASCAD, we observe causal effects on outputs by training probes on the final layer and selectively intervening on key components. However, further refinement is needed to extract mask values accurately. The experiments are presented in Appendix D.

## 6. Contextualizing the Results

In this work, we focus on side-channel models and attempt to find the features the trained networks extract by analyzing models after specific phase transitions during training. While several works have already explored the circuits in language models (Wang et al., 2023; Olsson et al., 2022) and those learned during phase transitions in algorithmic models (Nanda et al., 2023b; Simon et al., 2023), the feasibility of using these approaches for more realistic tasks remains an open question.

We showcase that studying phase transitions without a priori assumptions about relevant input features is still possible in real-world settings. Indeed, the SCA setting is fairly extreme in some characteristics: 1) the traces are extremely long and contain only small amounts of relevant information, 2) features can leak in several distinct ways, and 3) models are not expected to achieve high accuracies. As such,

the analysis requires investigating average/accumulated behavior across a large number of examples, and individual interventions become significantly more complex. We further showcase that deriving relevant features from outputs can be useful in determining model behavior.[9]

Our work shows a real-world example of effective reverse engineering of model predictions. We showcase that investigating phase transitions can be an effective approach to understanding the features networks learn, even in challenging settings. More precisely, we observe that phase transitions result in feature learning and corresponding generalization. Finally, the main generalizations of the models are all learned in discrete phase transitions, which are clearly detectable in both model performance and output logits as predicted by the quantization model (Michaud et al., 2023). We further showcase that some structures seem universal across different targets when the physical leakage is sufficiently similar, providing further evidence for weak universality (Chughtai et al., 2023). Indeed, the features learned after the first phase transition in both the ESHARD and CHES_CTF models are the same.

## 7. Implications of Results on the SCA Domain

As DLSCA becomes more common, it is increasingly important to understand how neural networks exploit implementations. This work provides concrete analyses for several common side-channel datasets, showing the possibility of reverse engineering masks from network activations. We show that specific structures can occur for different side-channel targets, indicating that building a library of such common attack paths or circuits can be useful in analyzing future networks.

As a highlight result, we can reverse engineer the secret shares from a trace by using the structures learned by the neural networks. To our knowledge, only (Gao et al., 2018)

---

[9]We also see the relevance of output-centric features for automated interpretability (Gur-Arieh et al., 2025).

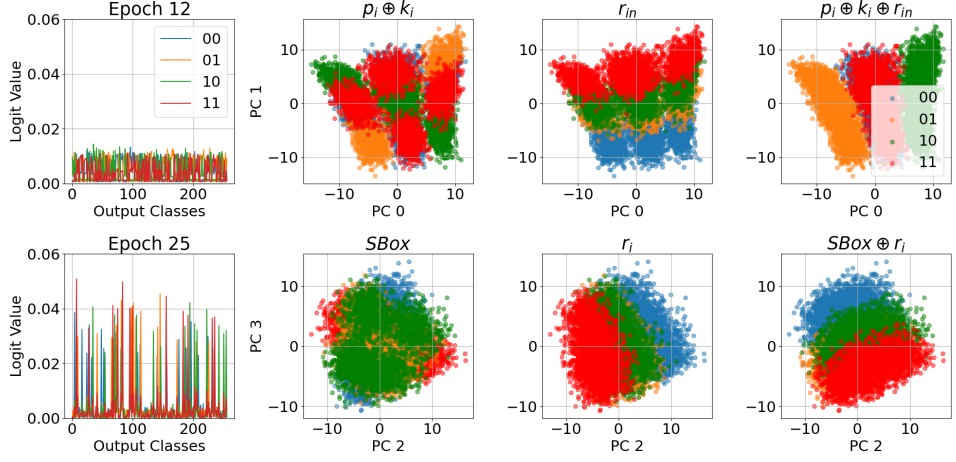

Figure 6: Logit analysis for two LSBs of $p_i \oplus k_i$ at epoch 12 and S-box$[p_i \oplus k_i]$ at epoch 25 with corresponding actual mask values for ASCAD. Note that for the lower logit plot, we use only traces with $p_i \oplus k_i$ in 00 for clarity, and that extracted mask values are in Figure 10.

can extract mask values, where this work is focused on a specific implementation using classical side-channel techniques (thus, no consideration of machine learning approaches), which requires stronger assumptions than DL-based attacks. This substantially benefits evaluations as we can move from black-box to white-box evaluations. This, in turn, would allow better feedback to designers of cryptographic implementations.[10] One might question how relevant this is for practical attackers as we require a model that already breaks the target. When attacks target individual bytes, the difficulty of breaking any individual byte can vary, even for the same device. As such, when masks are shared for all bytes (which is often required for masking the non-linear operations, e.g., the S-box in AES), spending significant effort to break one key byte might allow retrieving the shared mask. Then, subsequent attacks against other key bytes become much more straightforward as we can use the retrieved mask during training to effectively move the attack to an unprotected case by including the mask, see, e.g., the white-box evaluations in (Bronchain & Standaert, 2021).

Finally, discovering how neural networks practically defeat countermeasures can improve evaluation/attack methodologies and countermeasure design. On the evaluation/attack side, we can design more effective methods for label distribution that consider the common mistakes networks make, which can improve convergence (Wu et al., 2023). On the defense side, understanding what type of leakage is more/less easily exploited could lead to the design of more (cost-)effective countermeasures that enable more robust protections (Rijsdijk et al., 2022).

[10]See (Masure et al., 2023) for more discussion on the relevance in the context of SCA evaluations.

Profiled attacks against real-world targets are often significantly more complex than idealized evaluation settings where the same device is used for both profiling and attack. Differences between devices often result in worse performance when models trained on a profiling device are applied to the target (Bhasin et al., 2019). In security evaluations, the same device is commonly used for both profiling and attack to represent the worst-case scenario where the device differences are minimal. As such, the attack sets of the considered targets are from the same device as the profiling set, which raises questions about the practical relevance of these results for real-world settings. However, this work considers post-hoc explanations for models that already break a target. Therefore, the experimental evaluations emulate what is possible even for (more realistic) non-profiled adversaries that obtain a trained model using techniques like (Timon, 2019) as non-profiled attacks always consider a single device.

## 8. Conclusions and Future Work

We show that interpreting neural networks trained on side-channel models is feasible, even without access to random masks. Moreover, we highlight the effectiveness of investigating the structures learned during phase transitions and find evidence for the weak universality of circuits in side-channel models. Finally, we leverage these insights to reverse engineer the mask values.

Automating these analyses represents an interesting direction for future work. Additionally, further work on leveraging the insights into DLSCA models to improve evaluation methods could be useful. For example, using tailored leakage models that consider common structures could help simplify model tuning.

## 9. Impact Statement

This paper proposes a new approach to understanding the operations performed by neural networks when used in side-channel analysis. The end goal is to improve the security of implementations by implementing stronger countermeasures. As such, there are many potential societal consequences of our work, none of which we feel must be specifically highlighted here. We do not use live systems or violate terms of service, and to the best of our knowledge, we follow all laws. Our research does not contain elements that could potentially negatively impact team members. All used datasets are publicly available.

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

## A. Datasets

We utilize publicly available datasets commonly used in SCA literature for benchmarking. These datasets implement AES-128 with Boolean masking protection. The attack set consists of 10 000 traces for each dataset.

CHES CTF 2018 (Hu et al., 2019)[11] consists of power consumption measurements from an AES-128 implementation running on ARM Cortex-M4 (32 bits). CHES CTF 2018 raw traces contain 650 000 sample points per trace. Following (Perin et al., 2022b), we take a subset of 150 000 points corresponding to the initial setup and the first AES round and resample to 15 000 samples per trace. The profiling set has 30 000 traces.

ESHARD-AES128 (Vasselle et al., 2023)[12] consists of EM measurements from a software-masked AES-128 implementation running on an ARM Cortex-M4 device. The AES implementation is protected with a first-order Boolean masking scheme and shuffling of the `S-box` operations. In this work, we consider a trimmed version of the dataset that is publicly available [13] and includes the processing of the masks and all `S-box` operations in the first encryption round without shuffling. This dataset contains 100 000 measurements with 90 000 traces for the profiling set.

ASCAD (Benadjila et al., 2020) measures EM emissions from an AES-128 implementation on AVR RISC (8 bits). We use the version with the variable key in the profiling set. The traces are 250 000 sample points per trace. Following (Perin et al., 2022a), we take a window of 20 000 points, which are resampled to 2 000 points. 200 000 traces are used for profiling.

## B. Models and Training

The used models are MLPs from (Perin et al., 2022a), where model configurations were found through a random hyperparameter search for ESHARD and ASCAD. Note that as the ESHARD model performed well directly for CHES_CTF, we did not do further optimizations.

The model for CHES_CTF and ESHARD is a 4-layer MLP with 40 neurons in each layer with *he_uniform* weight initialization. We use *relu* activations. We use the Adam optimizer with a learning rate of 0.0025 and L1 regularization set to 0.000075. The batch size is 400, and we train for 200 epochs for CHES_CTF and 100 for ESHARD.

For ASCAD, the model is a 6-layer MLP with 100 neurons in each layer with *random_uniform* weight initialization. We use the Adam optimizer with a learning rate of 0.0005. We use *elu* activations, and we again train for 100 epochs with batch size 400.

## C. ESHARD Results

In Figure 7, we see that for ESHARD, only one phase transition occurs for the test set. At epoch 4, the perceived information becomes positive, and the models start to generalize. We note that the main distinction here is again the high-low HWs, similar to the first phase transition in CHES_CTF. Further analyses are analogous to CHES_CTF, although the model here can never distinguish between even and odd HWs.

In the rightmost two plots in Figure 8, we showcase distributions of the concrete intermediate values the models use. The models are clearly mapping the HWs of secret shares onto specific features.

### C.1. Activation Patching

We can do similar patching experiments as done for CHES_CTF in Section 5.1.1. As the high-low HWs are not on the diagonals in the PCs at epoch 5, we rotate the PC coordinates before patching and then rotate them back before continuing inference to align PCs more with the expected masks. The results we see in Figure 9 closely match the actual distributions of secret share HWs as seen in the rightmost plots of Figure 8.

---

[11] Referred to as CHES_CTF.
[12] Referred to as ESHARD.
[13] https://gitlab.com/eshard/nucleo_sw_aes_masked_shuffled

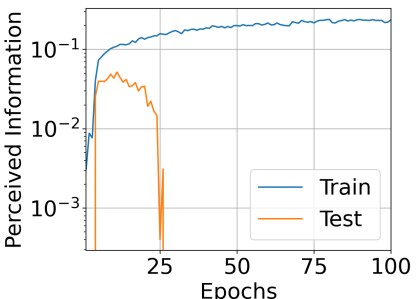

Figure 7: Evolution of Perceived Information for train and test traces of the ESHARD dataset.

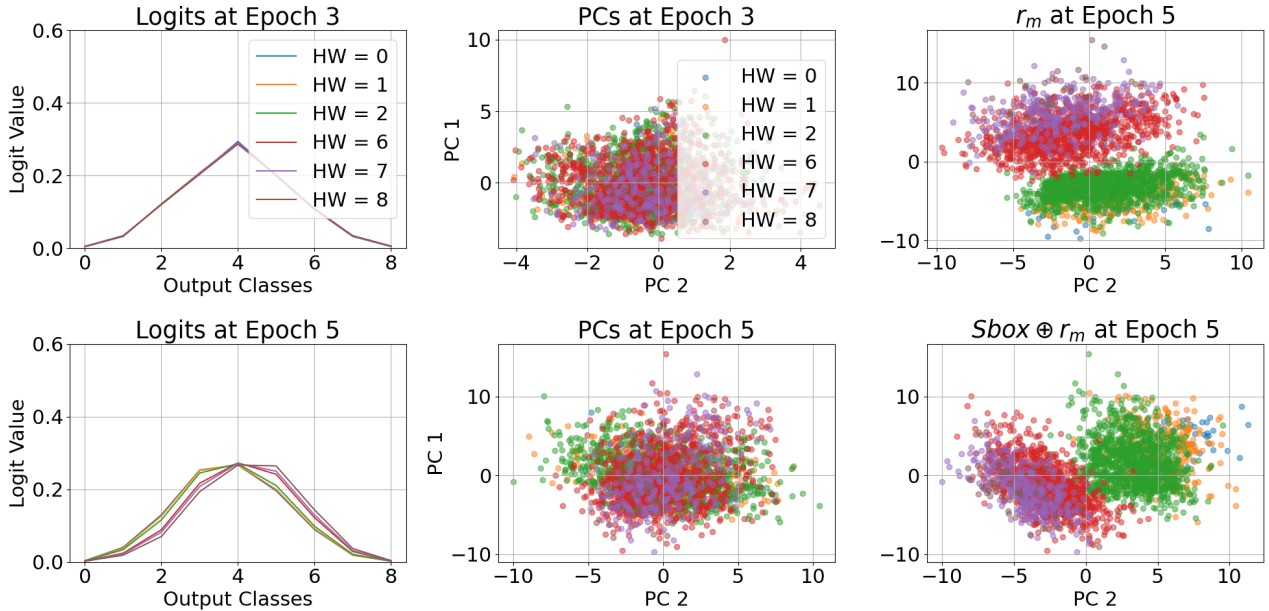

Figure 8: Logit analysis (first column) and activation analysis (second column) from models at epoch 3 (top row) and epoch 5 (bottom row). Legend is shared among all figures. We also include the PC embeddings for the actual mask of secret shares at epoch 5 (third column). The masks we extract are in Figure 9.

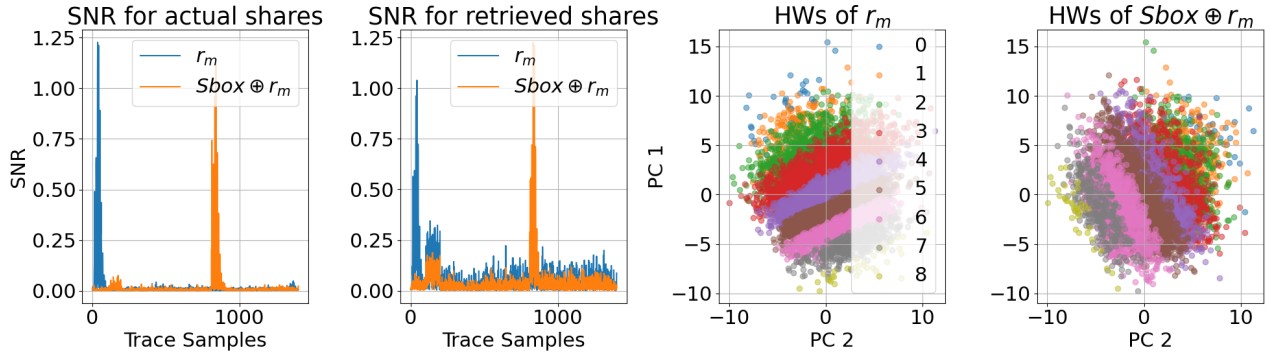

Figure 9: SNR plot and PC component distributions for mask values using patching experiments in ESHARD.

## D. ASCAD Patching results

As the leakage model for ASCAD is more complicated than the HW models, patching becomes more difficult. First, we train probes on the final layer to classify the input and output bits separately. We can directly measure the effects on only the input or output. Patching the input shares in the PC components in layer 2, which we show in Figure 6, does not work. Then, we find a qualitatively similar structure in PC1 and PC2 in layer one and patch there.

For the patches on the output shares, we set the first two components, which are related to the input shares, to 0 to isolate the effects of the patched components. For both experiments, we again rotate the two components by multiplying them with a rotation matrix to simplify the patches.

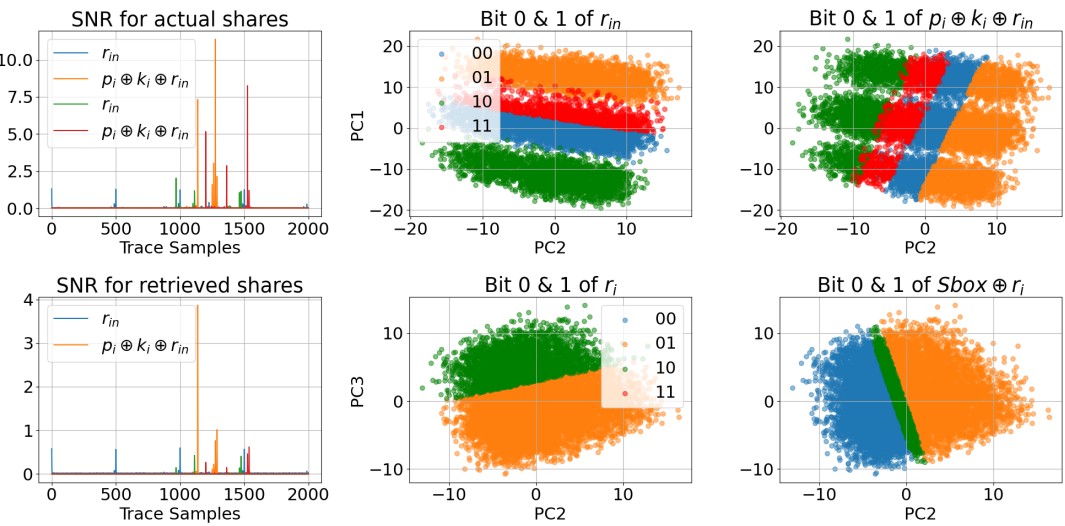

Figure 10: SNR plot and PC component distributions for mask values using patching experiments in ASCAD.

In Figure 10, we can see that the patches work reasonably well. Clearly, intervening on the found components has some causal effects. Furthermore, as we can see in the SNR plots, the patched outputs of the models are tied to the mask values we expect. However, the SNR values are significantly lower than those for the actual shares, and the $r_i$ and S-box $\oplus r_i$ shares only result in two or three classes, respectively, where we expect four. Additionally, we see that the reversed shares do not combine to the correct label for the input bits, indicating that while the mask values we retrieve are a reasonable clustering, further post-processing is necessary to retrieve actual values.

As we aim to keep the experiments (somewhat) aligned across all targets, we do not tailor the patching methods further for ASCAD. The current experiments show we can intervene in the structures and observe effects on the (probe) outputs. However, refining mask extraction methods in models with more complicated interactions is an interesting direction for future work. We provide further analysis to validate model predictions based on the four bits in Appendix F.1.

## E. HW Recombination CHES_CTF and ESHARD

Next, we discuss how mask recombination can be done algorithmically for the HW leakage model.

### E.1. High-Low HW Distinguishing

For the CHES_CTF and ESHARD targets, we notice that after the first phase transition (for some cases), high and low HWs can be differentiated. These are byte-based implementations protected with Boolean masking with order 2, i.e., the sensitive value $x = x_1 \oplus x_2$ ($\oplus$ being bitwise xor). When, based on prior experience working with these targets, we then choose to model the leakage (and therefore the presumed features of the model) as the $L = HW(x_i)$[14] we can consider modeling

---

[14]We note that we knew this a priori for ESHARD and it was strongly suspected for CHES_CTF. However, it is also a common leakage model in practice.

| $HW(x)$ | Matrices counting occurrences of $HW(s_1), HW(s_2)$ s.t. $x = s_1 \oplus s_2$ from 0-9. |
|---|---|
| HW = 0
HW = 1 | $\begin{bmatrix} 1 & 8 & 0 & 0 & 0 & 0 & 0 & 0 & 0 \\ 8 & 8 & 56 & 0 & 0 & 0 & 0 & 0 & 0 \\ 0 & 56 & 28 & 168 & 0 & 0 & 0 & 0 & 0 \\ 0 & 0 & 168 & 56 & 280 & 0 & 0 & 0 & 0 \\ 0 & 0 & 0 & 280 & 70 & 280 & 0 & 0 & 0 \\ 0 & 0 & 0 & 0 & 280 & 56 & 168 & 0 & 0 \\ 0 & 0 & 0 & 0 & 0 & 168 & 28 & 56 & 0 \\ 0 & 0 & 0 & 0 & 0 & 0 & 56 & 8 & 8 \\ 0 & 0 & 0 & 0 & 0 & 0 & 0 & 8 & 1 \end{bmatrix}$ |
| HW = 2
HW = 3 | $\begin{bmatrix} 0 & 0 & 28 & 56 & 0 & 0 & 0 & 0 & 0 \\ 0 & 56 & 168 & 168 & 280 & 0 & 0 & 0 & 0 \\ 28 & 168 & 336 & 840 & 420 & 560 & 0 & 0 & 0 \\ 56 & 168 & 840 & 840 & 1680 & 560 & 560 & 0 & 0 \\ 0 & 280 & 420 & 1680 & 1120 & 1680 & 420 & 280 & 0 \\ 0 & 0 & 560 & 560 & 1680 & 840 & 840 & 168 & 56 \\ 0 & 0 & 0 & 560 & 420 & 840 & 336 & 168 & 28 \\ 0 & 0 & 0 & 0 & 280 & 168 & 168 & 56 & 0 \\ 0 & 0 & 0 & 0 & 0 & 56 & 28 & 0 & 0 \end{bmatrix}$ |
| HW = 4 | $\begin{bmatrix} 0 & 0 & 0 & 0 & 70 & 0 & 0 & 0 & 0 \\ 0 & 0 & 0 & 280 & 0 & 280 & 0 & 0 & 0 \\ 0 & 0 & 420 & 0 & 1120 & 0 & 420 & 0 & 0 \\ 0 & 280 & 0 & 1680 & 0 & 1680 & 0 & 280 & 0 \\ 70 & 0 & 1120 & 0 & 2520 & 0 & 1120 & 0 & 70 \\ 0 & 280 & 0 & 1680 & 0 & 1680 & 0 & 280 & 0 \\ 0 & 0 & 420 & 0 & 1120 & 0 & 420 & 0 & 0 \\ 0 & 0 & 0 & 280 & 0 & 280 & 0 & 0 & 0 \\ 0 & 0 & 0 & 0 & 70 & 0 & 0 & 0 & 0 \end{bmatrix}$ |
| HW = 5
HW = 6 | $\begin{bmatrix} 0 & 0 & 0 & 0 & 0 & 56 & 28 & 0 & 0 \\ 0 & 0 & 0 & 0 & 280 & 168 & 168 & 56 & 0 \\ 0 & 0 & 0 & 560 & 420 & 840 & 336 & 168 & 28 \\ 0 & 0 & 560 & 560 & 1680 & 840 & 840 & 168 & 56 \\ 0 & 280 & 420 & 1680 & 1120 & 1680 & 420 & 280 & 0 \\ 56 & 168 & 840 & 840 & 1680 & 560 & 560 & 0 & 0 \\ 28 & 168 & 336 & 840 & 420 & 560 & 0 & 0 & 0 \\ 0 & 56 & 168 & 168 & 280 & 0 & 0 & 0 & 0 \\ 0 & 0 & 28 & 56 & 0 & 0 & 0 & 0 & 0 \end{bmatrix}$ |
| HW = 7
HW = 8 | $\begin{bmatrix} 0 & 0 & 0 & 0 & 0 & 0 & 0 & 8 & 1 \\ 0 & 0 & 0 & 0 & 0 & 0 & 56 & 8 & 8 \\ 0 & 0 & 0 & 0 & 0 & 168 & 28 & 56 & 0 \\ 0 & 0 & 0 & 0 & 280 & 56 & 168 & 0 & 0 \\ 0 & 0 & 0 & 280 & 70 & 280 & 0 & 0 & 0 \\ 0 & 0 & 168 & 56 & 280 & 0 & 0 & 0 & 0 \\ 0 & 56 & 28 & 168 & 0 & 0 & 0 & 0 & 0 \\ 8 & 8 & 56 & 0 & 0 & 0 & 0 & 0 & 0 \\ 1 & 8 & 0 & 0 & 0 & 0 & 0 & 0 & 0 \end{bmatrix}$ |

Table 1: Occurrences of HWs for two 8-bit shares for each of the nine output classes, cell $i, j$ in each matrix corresponds to $HW(s_1), HW(s_2)$. For the percentage of examples in practice, these values should be divided by $256^2$. As even and odd $HW(x)$ never occurs in the same place, we show two HWs in one matrix. Note that any red value (resp. black) is a zero in black (resp. red).

how occurrences of different classes $Y = HW(x)$ look. In Table 1, low HWs tend to be on the diagonal from top-left to bottom-right while high HWs tend to be on other diagonal. This (low HWs on one diagonal while high HWs are on the other) matches the PC embeddings for both models in Figure 8 and Figure 3.

### E.2. Even-Odd HW Distinguishing CHES_CTF

For CHES_CTF, we further see that the even and odd HW target classes can be distinguished after the second phase transition. From Table 1, it is clear that if the HWs of each secret share can be retrieved accurately enough, there should be a clear separation between even and odd HWs for the resulting point. Indeed, for any point $HW(x_1), HW(x_2)$ where $x = x_1 \oplus x_2$

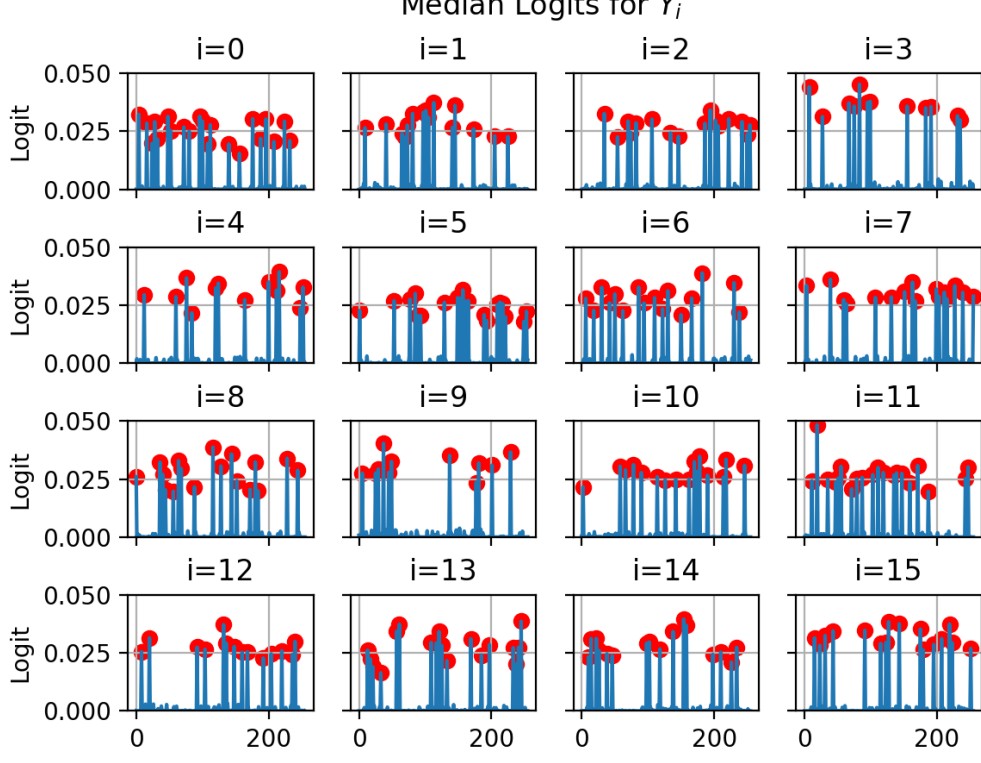

Figure 11: Median logits for traces belonging to varying $Y_i$ classes. The red dots indicate indices in respective $Y_i$.

we have that $HW(x_1) + HW(x_2) \mod 2 = HW(x) \mod 2$. This can be seen in Table 1 for two 8-bit shares, but the ability to distinguish the parity of $HW(x)$ holds for general higher masking orders $d$ (Ito et al., 2022).

## F. Bitwise Leakage ASCAD

As we show in Figure 5, the features the model learns for ASCAD are the two least significant bits of both $p_i \oplus k_i$ and S-box$[p_i \oplus k_i]$. We first note that the way the first two bits of S-box$[p_i \oplus k_i]$ relate to model labels (S-box$[p_i \oplus k_i]$) is straightforward: if bit 0 and 1 of S-box$[p_i \oplus k_i]$ are 00 then these correspond to predicting each label $y \mod 4 \equiv 0$. For bits 0 and 1 of $p_i \oplus k_i$, we can use the inverse of the S-box[15]. If we define $y' = $ S-Box$^{-1}[y]$ then if bit 0 and 1 of $p_i \oplus k_i$ are 00, we predict $y$ s.t. $y' \mod 4 \equiv 0$.

Combining these, we can divide the output classes into 16 clusters corresponding to model predictions. Practically, we define the outputs that belong to the 16 classes as $Y_i = \{y | y \equiv i \mod 4 \wedge y' \equiv \lfloor \frac{i}{4} \rfloor \mod 4\}$. Here, we set $i$ to be a concatenation of bit 1 and 0 of $p_i \oplus k_i$ and then bit 1 and 0 of S-box$[p_i \oplus k_i]$.

We can then train a linear probe on the activations of the final layer to predict these 16 classes. If we then transform the probe outputs to evenly distribute the predictions for its $i$'th output to the values in $Y_i$, we can measure the entropy between this resulting distribution and the model outputs. In summary, the probe accuracy is 0.64, and the PI between the probes' transformed distribution and the labels is 2.47 vs. 2.58 for the actual model. The entropy (in bits) between the probe outputs and model predictions is 0.27, indicating that most of the relevant behavior is explained by using the probe.

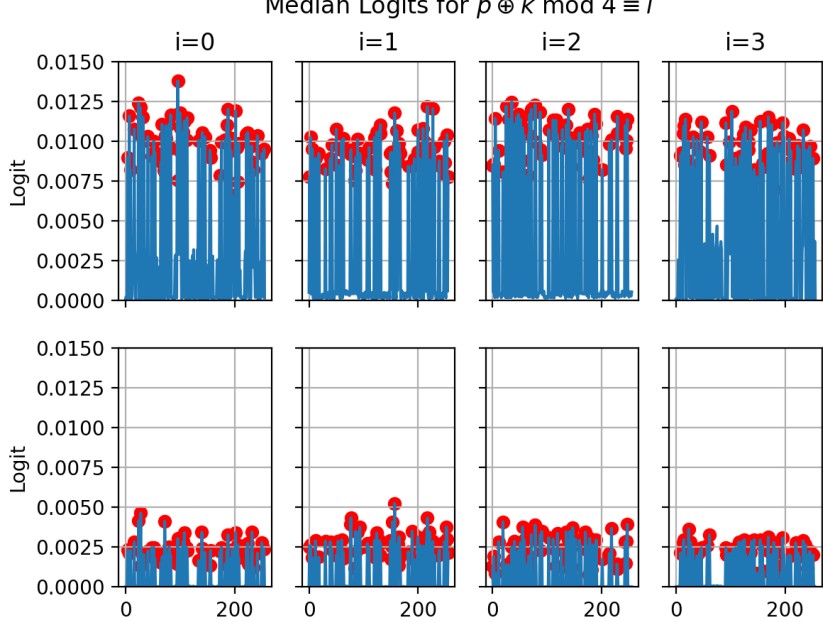

Figure 12: Median logits for traces belonging to varying for epoch 12 (top) and 25 (bottom).

### F.1. Logits For ASCAD with Classes

In Figure 11, we show the median logits for traces belonging to classes $Y_i$. As we can see, the logits corresponding to the expected points in $Y_i$ are always the main peaks.

Figure 12 shows how logits change from epoch 12 to epoch 25. When we analyze using only S-box inputs, we see that the logit values are significantly higher before accuracies for output bits are increased. This is explained by the fact that each of these cases combines four plots (vertically) in Figure 11. Concretely, as for each trace in Figure 12, we combine traces that belong to 4 different classes of the output bits, we expect the logits for each index that belongs to $p \oplus k \mod 4 \equiv i$ to only be high for 1/4 traces resulting in lower medians. Note that mean values do not show this same trend as the increase in the $Y_i$ class compensated for this decrease.

To verify that the results in Figure 11 are not an artifact of selecting traces, we visualize the same analysis for bits 2 and 3 in Figure 13. Clearly, the output values are significantly lower than for correct bits, indicating that these bits are (mostly) not being used by the model.

---

[15]The AES S-box is bijective, which simplifies this, but the analysis also works for surjective functions by taking the pre-image.

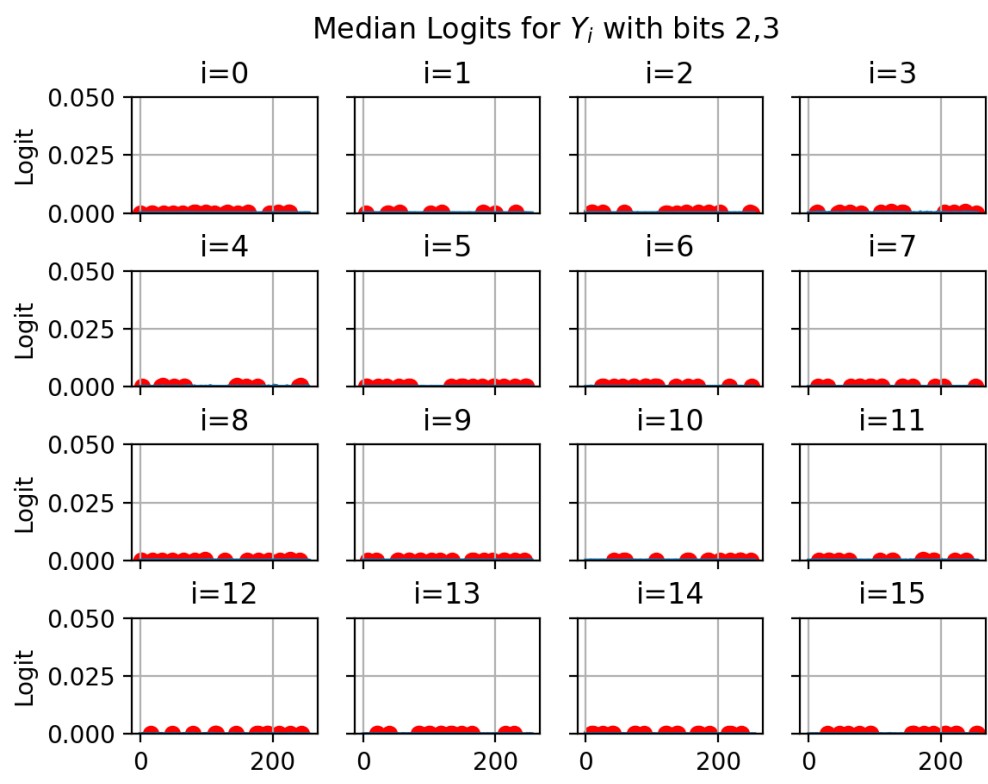

Figure 13: Median logits for traces belonging to varying $Y_i$ for bits 2 and 3. The red dots indicate indices in respective $Y_i$.

