# OpenReview forum: "It's Not Just a Phase: On Investigating Phase Transitions in Deep Learning-based Side-channel Analysis"
_ICML.cc/2025/Conference — Submitted to ICML 2025_

### Official Review · Reviewer_45qH · 2025-03-10

**Overall Recommendation:** 3

**Summary:**

This paper investigates deep learning-based side-channel analysis and introduces mechanistic interpretability methods to understand how neural networks trained on side-channel models learn. Specifically, the paper transforms black-box evaluation into white-box evaluation through reverse engineering, revealing the features learned by the network during phase transitions. The results on CHES_CTF, ESHARD, and ASCAD demonstrate the effectiveness of investigating the structures learned during phase transitions and provide evidence for the weak universality of circuits in side-channel models.

**Claims And Evidence:**

Yes. The claims made in the submission are well-supported by clear and convincing evidence.

**Essential References Not Discussed:**

No.

**Experimental Designs Or Analyses:**

Yes. Experiments on multiple datasets validate the effectiveness of the mechanistic interpretability analysis.

**Methods And Evaluation Criteria:**

Yes.  The Logit analysis method is used to identify key features and patterns in the model predictions. The activation analysis method is capable of finding physical leakage information in the principal components. The activation patching method is used to verify whether the features have causal relationships and is employed for reverse engineering the masks.

**Other Comments Or Suggestions:**

1.	For writing, it is recommended to adopt a general-to-specific structure, which would improve the clarity of the article. Specifically, start by introducing the overall framework before detailing the functions of individual modules.
2.	For the experimental analysis section, it is recommended to supplement the discussion with the impact of the main layers of the neural network, the training strategy, and the properties of the dataset.

**Other Strengths And Weaknesses:**

Strengths:
1.	This paper investigates the phase transition phenomenon in Deep Learning-based Side-channel Analysis.
2.	The motivation is clear, and the literature review is thorough.
3.	The phenomena revealed by the experiments are clear.
Weaknesses:
The paper lacks a discussion on the reasons behind the phase transitions. The main concern is about the role of network layers, the Adam learning rate, and the properties of the dataset itself.

**Questions For Authors:**

No.

**Relation To Broader Scientific Literature:**

Benadjila R, Prouff E, Strullu R, et al. Deep learning for side-channel analysis and introduction to ASCAD database[J]. Journal of Cryptographic Engineering, 2020, 10(2): 163-188.
Perin G, Karayalcin S, Wu L, et al. I know what your layers did: Layer-wise explainability of deep learning side-channel analysis[J]. Cryptology ePrint Archive, 2022.
This paper extends previous work by explaining deep learning side-channel analysis from a different perspective.

**Theoretical Claims:**

Yes. This paper does not involve theoretical statements. All hypotheses are based on existing literature, such as phase transition theory.

---

> ### Author Rebuttal · Authors · 2025-04-01
>
> Thank you for the review. We are glad the motivations and phenomena we describe are clear.
>
>
> W1: As mentioned in the paper, the (potential) reasons for learning occurring in discrete phase transitions are initially discussed in [1], offering preliminary insights into the phenomenon. More elaborate theoretical discussions on phase transitions can be found in the literature on Singular Learning Theory [2,3], which we consider out of scope for this work. However, we will add references relevant to these theoretical aspects to address the reviewer's concern.
>
> Other Suggestions:
> 1: We believe that our current structure already incorporates this principle. The introduction and the subsequent sections on SCA and MI are designed to provide a broad, general overview of the necessary background and concepts. These sections establish the foundation for understanding our approach, which is introduced and detailed in Section 4. We can highlight this further in our paper if it is not immediately apparent.
>
> 2: Our work has a section discussing the results and their implications in the SCA domain. We are not quite sure what you mean by `impact of the main layers', but if this concern is similar to the concerns raised by reviewer 6ugN (see weaknesses 2 and 4), please see the corresponding response. The dataset details and training process are provided in Appendices A and B.
>
> [1]: Michaud, E. J., Liu, Z., Girit, U., and Tegmark, M. The quantization model of neural scaling. NeurIPS 2023
>
> [2]: Watanabe, Sumio. Algebraic geometry and statistical learning theory. Vol. 25. Cambridge university press, 2009.
>
> [3]: Wei, Susan, et al. "Deep learning is singular, and that’s good." IEEE Transactions on Neural Networks and Learning Systems 34.12 (2022): 10473-10486.

---

> > ### Comment · Reviewer_45qH · 2025-04-05
> >
> > I double-checked the paper carefully and found that some of the key issues have already been addressed to some extent in the appendix and the rebuttal, such as the training strategy and the types of side channels in the dataset. As Reviewer 6ugN also noted, the role of network architecture—such as CNNs and MLPs—could benefit from further insights. While I still have some concerns in this regard, I believe they do not significantly affect the core contributions of the paper. Therefore, I have decided to raise my score by one point.

---

> > > ### Author Response · Authors · 2025-04-07
> > >
> > > Thank you for your careful re-evaluation of our paper and for acknowledging that some of your key concerns regarding the training strategy and side channel types have been addressed in the appendix and rebuttal.

---

### Official Review · Reviewer_6ugN · 2025-03-11

**Overall Recommendation:** 3

**Summary:**

The paper explores the novel concept of phase transitions within the context of Deep Learning-based Side-channel Analysis (DLSCA). It introduces an approach for mechanistic interpretability, aimed at understanding the detailed mechanisms of how deep learning models adapt and operate during the phase transitions associated with training, specifically targeting the field of side-channel analysis where sensitive data is at risk. The authors investigate these transitions to uncover the specific leakage points that DL models exploit, enhancing the transition from black-box to white-box understanding of model behaviors. This research reveals how networks adjust their internal representations and decision-making processes to improve attack performance, thereby offering insights into both enhancing attack strategies and developing robust countermeasures.

**Claims And Evidence:**

See strengths and weaknesses.

**Essential References Not Discussed:**

See strengths and weaknesses.

**Experimental Designs Or Analyses:**

See strengths and weaknesses.

**Methods And Evaluation Criteria:**

See strengths and weaknesses.

**Other Comments Or Suggestions:**

See strengths and weaknesses.

**Other Strengths And Weaknesses:**

Strengths:
1. Focuses on a lesser-studied aspect of DLSCA, providing fresh insights into the dynamic changes in model behavior during training, known as phase transitions. Offers deep insights into the internal workings of DL models, particularly how they handle and process side-channel data during phase transitions. Directly applies findings to improve methods for attacking cryptographic devices, highlighting practical applications in security.

2. Utilizes sophisticated techniques such as mechanistic interpretability to analyze the models, providing a higher resolution of understanding. Enhances the ability to perform white-box analyses of side-channel attacks, which is crucial for developing effective security measures.

3. Employs a comprehensive set of experiments that validate the theoretical findings, strengthening the claims with empirical evidence. Bridges gaps between deep learning, cryptography, and security analysis, appealing to a broad audience. By understanding how models learn during phase transitions, the research contributes to designing better countermeasures against side-channel attacks. The paper is well-written with detailed analyses that are both deep and accessible, providing clarity on complex concepts.

Weaknesses:
1. The advanced techniques used may be difficult to understand or implement without a deep background in both machine learning and cryptography.

2. The study might be overly tailored to the specific types of neural networks studied, which could limit generalizability. Assumes access to certain model insights that might not be available in more secure or differently configured systems.

3. The methods discussed may require significant computational resources, limiting their applicability in constrained environments. While the paper provides a robust approach to understanding DLSCA, it could benefit from comparing its methods against other possible analytical techniques.

4. The focus on specific types of side-channel datasets might not reflect the full range of scenarios where DLSCA could be applied. It is not clear how well the approaches discussed would scale to larger or more complex datasets and models. The effectiveness of the techniques may rely heavily on the quality and nature of the data used, which can vary significantly in real-world scenarios. There is a risk that the training process might introduce biases, which the phase transition analysis might not fully account for. The paper could provide a more detailed discussion on the conditions or scenarios where the proposed techniques fail to provide insights or improvements.

**Questions For Authors:**

See strengths and weaknesses.

**Relation To Broader Scientific Literature:**

See strengths and weaknesses.

**Theoretical Claims:**

See strengths and weaknesses.

---

> ### Author Rebuttal · Authors · 2025-04-01
>
> Thank you for the positive review. We are glad you found the analyses both deep and accessible.
>
> W1: While this might be true, we hope this initial work will simplify future analyses by enumerating some (potentially) common structures. Additionally, automating some of these analyses could be possible in future work to streamline the use of interpretability analysis. We can expand the discussion on this in Section 7.
>
> W2: While we only focus on MLPs in this work, the results on CNNs in [1] imply that these methods would translate there. Additionally, we see that MLPs from [2] and transformers from [3] learn the same algorithms for the same tasks, indicating consistency across architectures. We will add some more discussion on this to the camera-ready version.
>
> W3: The computational load for applying the methods in the paper is very low. The main computational load in this work is training the initial model, which was already done before applying our MI analysis and is the core part of conducting DLSCA. Since models in SCA are typically small compared to those from NLP or CV domains, training models for the considered targets generally takes under an hour on an RTX 4080 GPU.
>
> Alternative analytical techniques are either unsuitable or incomparable in this context. Many MI methods utilize input interventions that are impossible for the DLSCA analysis (as discussed in the paper), and the input visualization techniques previously used in SCA do not target model internals. Lastly, [1] requires access to masks, which makes it not directly comparable.
>
> W4: Indeed, analyzing more difficult targets will be more challenging. However, we want to emphasize that our dataset selection already includes a range of complexity, with ESHARD being significantly noisier than CHES_CTF. However, we recognize this as a critical area for future investigation.
>
> Our analysis objective here is to understand what the model has learned rather than what it should have learned. The introduction of biases during training is indeed a concern. If a model fails to capture certain leakage during training, our analysis will not reveal them. However, classical SCA approaches often rely on significant assumptions about the targets, introducing their own biases, while (black-box) DLSCA potentially mitigates having to make these assumptions. We will expand the discussion in Section 7.
>
>
> [1]: Perin, G., Karayalcin, S., Wu, L., and Picek, S. I know what your layers did: Layer-wise explainability of deep learning side-channel analysis. Cryptology ePrint Archive, Paper 2022/108
>
> [2]: Chughtai, B., Chan, L., and Nanda, N. A toy model of universality: Reverse engineering how networks learn group operations. In Krause, A., Brunskill, E., Cho, K., Engelhardt, B., Sabato, S., and Scarlett, J. (eds.), International Conference on Machine Learning, ICML 2023
>
> [3]: Nanda, N., Chan, L., Lieberum, T., Smith, J., and Steinhardt, J. Progress measures for grokking via mechanistic interpretability. In The Eleventh International Conference on Learning Representations, ICLR 2023

---

> > ### Comment · Reviewer_6ugN · 2025-04-08
> >
> > Thanks for the author's rebuttal. The author's response did not solve all my questions well, so I kept my previous rating.

---

### Official Review · Reviewer_kqw9 · 2025-03-11

**Overall Recommendation:** 3

**Summary:**

The paper applies mechanistic interpretability techniques to side-channel analysis, which is used to extract secret keys from protected devices by monitoring physical factors like power consumption. The authors investigate model behavior during phase transitions (sudden jumps in accuracy) by analyzing activations in MLP models. By projecting layer activations to 2D space using PCA and color-coding by target values, they identify interpretable patterns in learned representations. They validate these findings by altering the identified components and observing predictable changes in model output, demonstrating they’ve identified the causal mechanisms behind the model’s predictions.

## update after rebuttal
I have updated my score to 3, as the issues I previously mentioned have been resolved. However, the paper could still benefit from further clarifying its claim regarding tracing learned features back to specific input characteristics—particularly, what is meant by "input characteristics." - does this refer to internal device values like HW, or to model inputs like device temperature.

**Claims And Evidence:**

Yes

**Essential References Not Discussed:**

Not found.

**Experimental Designs Or Analyses:**

They are all sound.

**Methods And Evaluation Criteria:**

Yes

**Other Comments Or Suggestions:**

None

**Other Strengths And Weaknesses:**

## Strengths
1. The research experimented on multiple datasets and attack settings, showing robustness of their approach.
2. They successfully apply interpretability techniques to a real-world security problem. It is a potentially valuable test bed for future mechanistic interpretability research.
3. This method uniquely handles situations where direct input interventions aren’t possible.

## Weaknesses
1. The paper doesn’t fully explore how their findings could be used to improve device security, despite this being one of their stated motivations.
2. There’s no analysis connecting the learned features back to specific input characteristics that cause the leakage.
3. Their reliance on PCA indirectly assumes features are orthogonal, which may not hold true due to feature superposition effects documented in recent research.

**Questions For Authors:**

None

**Relation To Broader Scientific Literature:**

The paper connects two fields: mechanistic interpretability and side-channel analysis security research. In mechanistic interpretability, their approach of investigating activations based on output classes offers a novel technique for understanding model behavior, especially when direct input interventions aren’t possible.

**Theoretical Claims:**

No theoretical claims.

---

> ### Author Rebuttal · Authors · 2025-04-01
>
> Thank you for the feedback. We are pleased you think DLSCA might be a valuable testbed for future MI research.
>
> W1: Improving the security of devices requires a good understanding of the devices' vulnerabilities, and this work provides a concrete approach to understanding how and why a particular attack was successful. Additionally, this work offers several insights into the training dynamics in DLSCA, which can aid chip designers in utilizing countermeasures that aim to make these structures more challenging to learn. The recently introduced prime-field masking approach from [1] seems a viable approach. We will add some more discussion on this.
>
> W2: The SNR plots with reverse-engineered masks tie the extracted masks back to specific input points. Furthermore, the learned structures within the model provide insights into how the input values leak (HW for ESHARD/CHES and 2 LSBs for ASCADr). Therefore, we want to emphasize that our analysis does connect learned features to specific input characteristics that cause leakage, but we will highlight these points more in the paper.
>
> W3: The reviewer makes a valid point regarding the potential limitations of relying on PCA. However, we argue that the characteristics of the DLSCA models mitigate this limitation. As discussed in Section 4 (Activation Analysis) and supported by experimental findings using information bottlenecks in [2], DLSCA models typically learn a relatively small number of features. This reduces the likelihood and impact of significant feature superposition.
> On the other hand, alternative techniques could be useful and offer potential improvements if there is significant feature superposition, such as sparse autoencoders [3].
>
> [1]: Cassiers, G., Masure, L., Momin, C., Moos, T., & Standaert, F.-X. (2023). Prime-Field Masking in Hardware and its Soundness against Low-Noise SCA Attacks. IACR Transactions on Cryptographic Hardware and Embedded Systems, 2023(2), 482-518. https://doi.org/10.46586/tches.v2023.i2.482-518
>
> [2]: Perin, G., Karayalcin, S., Wu, L., and Picek, S. I know what your layers did: Layer-wise explainability of deep learning side-channel analysis. Cryptology ePrint Archive, Paper 2022/108
>
> [3]: Bricken, et al., "Towards Monosemanticity: Decomposing Language Models With Dictionary Learning", Transformer Circuits Thread, 2023.

---

> > ### Comment · Reviewer_kqw9 · 2025-04-02
> >
> > Thank you for your rebuttal. Your answers to W1 and W3 are both convincing.
> >
> > I believe the second issue—connecting learned features back to specific input characteristics—could be further clarified. Input characteristics can be understood in two distinct ways:
> > 1. Device internal features, such as the Hamming Weight (HW) or 2 LSBs (which I assume this is what you mean).
> > 2. Model inputs, which consist of side-channel traces, such as power consumption or electromagnetic (EM) signals.

---

> > > ### Author Response · Authors · 2025-04-07
> > >
> > > Thank you for the quick reply!
> > >
> > > On the input characteristics, the two categories you mentioned are indeed broadly what we care about from the SCA side.
> > >
> > > 1. More precisely, we are concerned with how the values manipulated by the (AES) algorithm running on the device leak (i.e., how the processed values influence the power usage or EM emissions). This is impacted by device internals and algorithm implementation but can also be influenced by the measurement setup and other external factors. In the SCA literature, the way the processed values leak in the measured side-channel information is referred to as the leakage model, such as the mentioned HW and LSB. In security evaluation or attack scenarios, these leakage models are often hypothesized. However, our results indicate that we can deduce this leakage model (or, more precisely, what leakage the model extracts). This is the Hamming weight (HW) for CHES_CTF and ESHARD and the two least significant bits for ASCADr.
> > >
> > > 2. Considering the side-channel traces (model inputs), we try to understand where those values are manipulated during (AES) computation. A common method for determining where a value is in a trace is to compute the signal-to-noise ratio (SNR) (or correlation) between the values and each point in the trace across many traces. By doing this with the extracted masks, we can show where each of these masks leaks within the trace (see Figure 4 (left)). By looking at this SNR plot, we can then also disambiguate which of the extracted values is the mask $r_m$ and which is the masked Sbox output $Sbox \oplus r_m$, as we can assume the masks have to be loaded before the masked Sbox computation can occur.
> > > We will add a section/paragraph to the discussion to clarify the above. We will also expand on the SNR description for better understanding.

---

### Official Review · Reviewer_Vb6J · 2025-03-14

**Overall Recommendation:** 3

**Summary:**

This paper investigates the feasibility of applying Mechanistic Interpretability (MI) to deep learning-based side-channel analysis (DLSCA) to enhance the interpretability of deep neural networks in security evaluations. The authors explore how neural networks exploit side-channel leakage and identify learned structures during training phase transitions. They also successfully demonstrate that networks extract secret mask values by analyzing model logits, principal components, and activation patching, effectively transitioning the evaluation from black-box to white-box.

**Claims And Evidence:**

Yes.

**Essential References Not Discussed:**

No.

**Experimental Designs Or Analyses:**

The experimental designs look good to me in genereal. However, I am a bit confused by what is the type of side channels of the dataset. I checked Appendix A and saw that the dataset contains execution traces of programs. Are these traces related to timing, cache, or even power side channels?

**Methods And Evaluation Criteria:**

I appreciate this work; while I am not familiar with Mechanistic Interpretability, I could understand how the features of deep learning-based side-channel analysis are learned and the key structures are identified. I only have one conce: what is the uniqueness of the investigation of applying MI to DLSCA? Could that be possible for the proposed method to be extended to other security related tasks, e.g., buffer overflow detections. If so, could the authors briefly explain the possible future direction? If not, could the authors justify why the proposed method is only applicable to side channel analysis?

**Other Comments Or Suggestions:**

N/A.

**Other Strengths And Weaknesses:**

### Pros

- Interesting and impo topics.
- The paper tries to enhance the interpretability of DLSCA.

### Cons

- The types of side channels in the dataset are not clearly introduced.

- Lack discussion to apply this method to other simialr fields.

**Questions For Authors:**

- What is the uniqueness of the investigation of applying MI to DLSCA? Could that be possible for the proposed method to be extended to other security related tasks?

- What are the side channel types of the execution traces used in the dataset?

**Relation To Broader Scientific Literature:**

The contribution of this paper is concrete and could be able to enhance the future research on using Deep Learning for SCA.

**Theoretical Claims:**

I checked the analysis approaches in Section 4, including Logit Analysis, Activations Analysis, and Reverse engineering masks with activation patching. These claims look good to me.

---

> ### Author Rebuttal · Authors · 2025-04-01
>
> Thank you for the feedback. We are glad to hear the core concepts were understandable, even without extensive prior knowledge of MI or DLSCA.
>
> Q1: The principles of MI can indeed be extended to other security-related tasks, but the specific analysis techniques and challenges can vary significantly across domains. Our focus on DLSCA was motivated by the unique challenges it presents for MI, as it involves noisy and complex data, and we cannot perform well-defined input interventions. Moreover, we observe that the number of phase transitions in DLSCA models is small, allowing us to perform a detailed, individual examination of each transition, as the number of features relevant for classification is relatively small (see W3 for reviewer kqw9).
> Thus, the specific techniques and the scale of the MI analysis must be adapted to the unique characteristics of the task and will (presumably) require significant expertise in those tasks. This work on DLSCA (and other related work) provides a valuable foundation for exploring these adaptations in future research.
>
> We also note that for security-related tasks, network performance alone is often not sufficient for wide adoption. In tasks like malware detection [1] or fraud detection [2], using classification algorithms (without specific explanations) in production might not be allowed for legal reasons. Similarly, for NNs that attack cryptography (either using DLSCA or more ML-based differential cryptanalysis [3]), a pass/fail condition is only of limited use without explanations. We will add some more discussion on the broader uses of MI in security-related applications.
>
> Q2: Utilized datasets consist of power (CHES_CTF) and electromagnetic emission (ESHARD and ASCAD) measurements from cryptographic executions on embedded devices (e.g., 15,000 points of power use across the first round of AES for CHES_CTF). This information is already stated in Appendix A, but we will mention it in the main text to clarify further.
>
>
> [1]: Saqib, Mohd, et al. "A Comprehensive Analysis of Explainable AI for Malware Hunting." ACM Computing Surveys 56.12 (2024): 1-40.
>
> [2]: Parkar, Erum, et al. "Comparative study of deep learning explainability and causal ai for fraud detection." International Journal on Smart Sensing and Intelligent Systems 1 (2024).
>
> [3]: Gohr, Aron. "Improving attacks on round-reduced speck32/64 using deep learning." Advances in Cryptology–CRYPTO 2019: 39th Annual International Cryptology

---

### Decision · Program_Chairs · 2025-05-01

**Decision:**

Reject

**Comment:**

This paper applies interpretability techniques to trained models in deep learning-based side-channel analysis (DLSCA), uncovering how networks internally represent cryptographic mask values. The topic is timely and technically interesting, and the work attempts to bridge concepts from machine learning theory with real-world cryptographic evaluation. After a thorough discussion and in agreement with the reviewers and the senior area chair, I recommend rejection.  The core insight—that models not only break the target but reconstruct internal secret shares—is well-demonstrated. However, the current presentation would benefit from clearer framing. The methodology relies on models that already succeed at key recovery, which narrows its scope to post hoc explanation and should be made more explicit. While the analyses are convincing, they involve a significant amount of manual tuning and dataset-specific intuition; it would strengthen the paper to more clearly acknowledge this and discuss paths toward broader applicability or automation. Concerns remain regarding the clarity and accessibility of the presentation. No reviewer expressed strong support for acceptance, even after rebuttal. Overall, while the paper touches on a relevant intersection of fields, it does not currently meet the standards of clarity, novelty, and accessibility expected at ICML.

The following issue was raised in the discussion, which did not contribute to the decision, but the authors should consider: The repeated use of “phase transition” (including the title) to describe sudden changes in training dynamics is misleading and detracts from the otherwise sound analysis. The analysis is based on observed sharp changes in performance.  These may be abrupt, but they lack the formal ingredients of a true phase transition (e.g., order parameters, asymptotic limits, non-analyticity of free energies in the large size limit). Reframing these as "sudden generalization" or "feature emergence" would more accurately reflect what is shown, and help focus the reader’s attention on the interpretability results, which are the true contribution of the paper.